# Detecting and Responding to Concept Drift in Business Processes

Lingkai Yang [1,*], Sally McClean [1], Mark Donnelly [1], Kevin Burke [2] and Kashaf Khan [3]

1 School of Computing, Ulster University, Jordanstown BT37 0QB, UK; si.mcclean@ulster.ac.uk (S.M.); mp.donnelly@ulster.ac.uk (M.D.)
2 MACSI, Mathematics Applications Consortium for Science and Industry, University of Limerick, Limerick V94 T9PX, Ireland; kevin.burke@ul.ie
3 British Telecom, Ipswich IP5 3RE, UK; kashaf.khan@bt.com
* Correspondence: yang-l9@ulster.ac.uk

**Abstract:** Concept drift, which refers to changes in the underlying process structure or customer behaviour over time, is inevitable in business processes, causing challenges in ensuring that the learned model is a proper representation of the new data. Due to factors such as seasonal effects and policy updates, concept drifts can occur in customer transitions and time spent throughout the process, either suddenly or gradually. In a concept drift context, we can discard the old data and retrain the model using new observations (sudden drift) or combine the old data with the new data to update the model (gradual drift) or maintain the model as unchanged (no drift). In this paper, we model a response to concept drift as a sequential decision making problem by combing a hierarchical Markov model and a Markov decision process (MDP). The approach can detect concept drift, retrain the model and update customer profiles automatically. We validate the proposed approach on 68 artificial datasets and a real-world hospital billing dataset, with experimental results showing promising performance.

**Keywords:** business process; concept drift; duration drift; hierarchical Markov model; Markov decision process



## 1. Introduction

A business process is a collection of interconnected tasks aiming to provide a product or service to customers [1]. Every business process generates data as a result of its activities, which is extremely valuable in customer understanding, process optimization and product qualification if properly analyzed [2]. Process mining (PM) is a research area that connects data mining and process modelling [2]. PM has a wild range of techniques to discover process models [1], monitor customer behaviour [3], spot business bottlenecks [4], and detect concept drift [5], etc. Concept drift, also known as process drift in a PM context, refers to business changes over time, which can be detected from various perspectives, such as control-flow [1,6], data-flow [7], resource [8] and time [9,10]. Detecting such changes can provide insight into business circumstances, generate early-stage change warnings and highlight the opportunity for model refinement [5].

Process control-flow and data-flow are well addressed by business process management (BPM) techniques, with the support of database technology [11]. However, less attention has been devoted to other perspectives such as resource management [11]. Two other literature review papers report similar findings. R'bigui and Cho [3] reviewed the research progress in tackling PM challenges outlined in the Process Mining Manifesto [2]. Regarding concept drift, ten of the eleven papers published between 2014 and 2017 only considered changes from a control-flow perspective. The remaining one discussed two perspectives: data and resources. Elkhawaga et al. [12] conducted a comprehensive study using nineteen papers published between 2009 and 2019. A conclusion is given that twelve of them were solely focused on changes in the control-flow structure and the other seven

publications took into account more than two perspectives such as time and resource. From a time perspective, concept drift is indicative of the variation in the time point when events occur, as described in [12]; however, it can also refer to changes in the time duration between events, which has not been well-investigated [3,12].

To fill the research gap, this paper proposes a hierarchical Markov model-based Markov decision process (HMDP) method for modelling customer duration data, detecting concept drifts and automatically updating customer profiles. Businesses can gain benefits from automating their processes, such as saving employees' time, reducing costs and increasing efficiency [13], which is important in application areas, for example, for Test-Driven Data Analysis (TDDA) [14] and recommendation systems [15]. The idea of modelling the concept drift detection problem as a Markov decision process (MDP) originates from Liebman et al.'s work [16], which aims at solving feature-based supervised learning and distribution tracking problems. While in this paper, we concentrate on the PM context and propose a hierarchical Markov model (hierarchical MM) approach for process modelling. Specifically, in our previous work [17], we explored the possibility of detecting customer duration drift by using a semi-Markov process (SMP) model. In this paper, we applied it in our hierarchical MM (the bottom layer) and proposed a corresponding MDP stage to autonomously respond to concept drift. In the data modelling stage, the proposed HMDP approach represents customer behaviour using a two-layer hierarchical MM. The bottom layer is an SMP integrated with gamma mixture models to characterize customer transition probabilities and duration. The top layer, derived from the K-means clustering, is a high-level representation of the bottom layer customer behaviour. In the drift adaptation stage, the method first characterizes the old and new data separately using the hierarchical MM. The drift adaptation, i.e., customer profile updating problem, is then considered as an MDP. Specifically, the top-layer customer representations of the old and new data are combined as the MDP states where actions occur, i.e., keep (no drift), adapt (gradual drift) or retrain (sudden drift) the old hierarchical MM.

This paper makes three main contributions. First, we represent the business process as a hierarchical MM, with customer activities and their duration as the bottom layer and customer performance patterns as the top layer. The aim of introducing the top layer is to reduce the difficulty of establishing the MDP model. Considering a process with $n$ activities, the bottom layer representation (i.e., the SMP model) has $n^2 + (3m - 1) * n^2$ parameters. The first item (i.e., $n^2$) refers to the number of transition probabilities. $(3m - 1) * n^2$ relates to the number of parameters in the gamma mixture model (with $m$ gamma components) of each of the $n^2$ transitions. A large number of parameters can lead to a huge MDP state space, making model training more difficult and demanding more computational resources [18]. By introducing the top layer, the number of parameters can be reduced significantly, depending on the number of predefined clustering centres in K-means. As a result, it can avoid the explosion of the MDP state space. Second, the customer duration data in the bottom layer is modelled using gamma mixture models. There are two main motivations: (1) when properly parameterized, the gamma mixture model is extremely flexible to fit any shape of continuous distribution with non-negative values [19]. (2) The mixed gamma class has been thoroughly investigated, including mathematical theory and proofs that are useful for model explanation and data comprehension. The third contribution is, that we consider drift detection and customer behaviour modelling as an MDP/reinforcement learning problem, with three actions to keep, adapt or retrain the old hierarchical MM to ensure that it is a sufficient representation of the dynamic business environment. In conclusion, this paper addresses the importance of the modelling process duration data, as well as adapting/updating customer profiles to the dynamic environment.

The rest of the paper is organized as follows. Section 2 presents the related work. Preliminaries used in this paper are given in Section 3. The methodology of the proposed approach is introduced in Section 4. The findings from a series of experiments on artificial and real-world datasets are presented in Sections 5 and 6, respectively. Finally, conclusions and the scope for future work are provided in Section 7.

## 2. Related Work

### 2.1. Process Mining

Concept drifts refer to data changes over time, which have been classified into four categories in the literature [20]. Sudden drift occurs abruptly at a specific time point with one model replaced by another one, while in a gradual drift scenario, the current model is taken over by a new model gradually. The drift type of recurring concept relates to situations in which previously seen concepts reoccur after some time. Incremental drift refers to a sequence of changes to reach a stable new model from the current model. In this paper, we consider recurring and incremental drift as special types of sudden and gradual drift, respectively [20,21].

In the context of process mining (PM), concept drift detection techniques can be roughly classified into online and offline learning approaches [22]. The former concentrate on new incoming data batches, with the objective to quickly respond to changes [22]. The latter detectd concept drift in history data, aiming for accurate data modelling and understanding [16]. Ostovar et al. [23] introduced an online approach aiming at detecting changes in its early stage. Eravolo et al. [24] reviewed the current state of online process mining, focusing on evaluation goals and event logs for drift detection. In [25], Adams et al. discussed the significance of understanding the root causes of concept drift. Sliding windows are commonly applied in offline scenarios to generate continuous populations [1,5,26] where changes are examined between consecutive pairs. The follows/precedes relationship, introduced by Bose et al. [1], investigates whether process activity 'a' always, sometimes or never follows activity 'b'. Thereafter, Hotelling $T^2$, KS and MW tests were employed to uncover process drift. Maaradji et al. [26] restructured process instances into RUNs as the representation of customer pathways. The Chi-square test is then used to detect changes. Clustering methods such as AHC [27], Markov clustering [28] and DBSCAN [29] are also commonly employed for process drift detection. Yeshchenko et al. [24] proposed the Visual Drift Detection (VDD) approach with the capability of users visually observing drift.

In addition to drift detection, concept drift adaptation is essential to ensure that the learned model/customer profile is suitable for the dynamic environment [21]. Junior et al. [30] proposed a framework for monitoring concept drift in process event streams. The method first represents process data into a trace histogram and a time histogram. Process cases can then be transformed into feature vectors to obtain the cluster boundary. New process cases that fall outside the boundary of any cluster are detected as anomalies. Finally, concept drift is considered to occur if there is a significant increase in the number of anomalies. The approach is then extended for real-time reaction and concept drift adaptation [31]. The method addresses drifts in the time difference/duration between activities but does not distinguish the difference between different activity transitions. Spenrath and Hassani [32] proposed a concept drift adaption method with the goal of finding process bottlenecks in cases that have a long duration. The training process cases are partitioned into a set of groups to build an ensemble-based drift detection model. The approach can adapt to gradual and recurrent concept drift, but it only concentrates on the overall duration, neglecting the duration of specific transitions. In [21], Maisenbacher et al. proposed a method for dealing with concept drift in predictive process monitoring. The approach uses incremental learning to adapt the forecast model to control flow and data concept drift, but there is a lack of duration perspective.

The proposed HMDP approach can automatically adapt to concept drift, i.e., update customer profiles under changes in customer transition probabilities and their duration. The method brings about two main benefits: (1) it models the duration distribution of each transition to highlight variations in the time spent between different activities. (2) The duration data of a specific transition is fitted using a gamma mixture model, which is more flexible and comprehensive than the histogram or the average value-based methods.

### *2.2. Markov Models*

A discrete-time Markov chain (DTMC) is usually used to characterize a stochastic process, in which the next state depends only on the current state with transition time/sojourn time/waiting time as one unit time [18]. A continuous-time Markov chain or a semi-Markov process (SMP) is commonly employed if the transition time is exponentially or randomly distributed [33]. SMP models are widely used for, e.g., modelling highway bridge maintenance [34], characterising patient pathways via hospital care [35] and optimising mode control [36]. The model also suits business process mining, but it has not been studied much. In general, a business process can be naturally represented in terms of an SMP framework by considering process activities as the state space, describing the probability of moving to the next activity from the current by transition probabilities and simulating the transition time by non-exponential distributions. It offers a single model for characterizing the process control-flow and the time spent by customers. A hierarchical MM is a stochastic process including at least two layers that usually follow the Markov property. Ferreira et al. [37] proposed a hierarchical MM to discover the relationship between low-level process activities and the high-level description of business processes. Mitchell et al. [38] integrated the Coxian phase-type model with the continuous-time Hidden Markov model to describe the length of stay of patients in hospital.

A Markov decision process (MDP) is a discrete-time stochastic process aiming to model decision-making in situations [18,39]. That is, an MDP is a DTMC with a decision/action selection policy in every transition. MDP can be solved by using dynamic programming-based approaches [39], which require updating the transition probability matrix and the transition reward matrix for every decision, leading to the cost of a substantial amount of computational resources [18]. Reinforcement learning (RL), especially approximation-based methods that estimate the value of state-action pairs (e.g., Deep Q network) [40], can be considered as a near-optimal solution for large and complex MDP problems [41]. On the other hand, such methods lack the ability to model transition probabilities of moving from the current to the next state via a selected action. MDP has been successfully applied in a wide range of action selection areas such as robotics and control [42], games [43], trading [44] and resource allocation [11,41], but it has received less attention in PM.

### 3. Preliminaries

This section introduces the formal preliminaries used throughout the paper, including details for the business event log, SMP, MDP and the gamma mixture model.

### *3.1. Business Process Event Log*

**Definition 1** (Event log)**.** *An event log L is a collection of recorded events with their attributes relating to process executions/activities. Typical attributes include the case ID, activity, timestamp, and resource, etc. Events with the same case ID consist of a process instance.*

An example of a hospital billing log is given in Table 1, illustrating two specific cases with the process control-flow structure demonstrated in Figure 1.

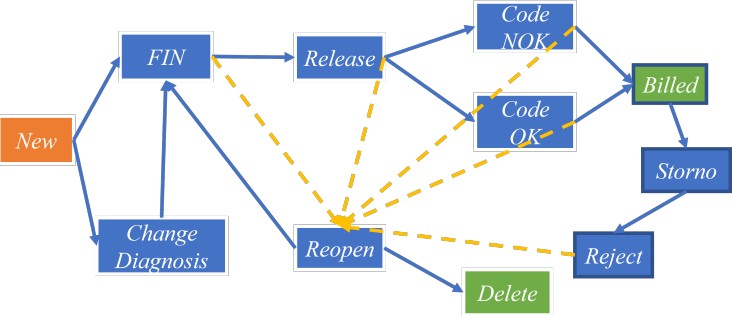

**Figure 1.** The hospital billing process.

**Table 1.** An example event log.

| Case ID | Activity | Timestamp |
|---|---|---|
| case0 | New | 11 February 2015 16:30:00 |
| case0 | FIN | 11 February 2015 16:30:00 |
| case0 | Release | 11 February 2015 09:59:02 |
| case0 | Code OK | 12 February 2015 09:59:02 |
| case0 | Billed | 12 February 2015 10:21:01 |
| case1 | New | 11 May 2015 12:07:00 |
| case1 | Change Diagnosis | 12 May 2015 12:30:40 |
| case1 | Release | 17 May 2015 12:54:09 |
| case1 | Code OK | 21 May 2015 15:22:14 |
| case1 | Billed | 12 June 2015 09:21:29 |
| . . . | . . . | . . . |

*3.2. Semi-Markov Process (SMP)*

As a generalization of DTMCs, SMPs allow state transitions to occur at continuous irregular times. We begin with some definitions that can be found in [33,45] to formulate an SMP framework. In general, an SMP can be represented by the three-tuple $\langle S_S, P_S, T_S \rangle$ referring to the state space, transition probabilities and transition time distributions.

**Definition 2 ($S_S$).** *The state space is used to describe a system/problem which can be finite (e.g., the number of people in the queue in a supermarket) or near-infinite (e.g., camera images in an automatic driving application). In a PM context, we consider the state space is finite.*

**Definition 3 ($P_S$).** *The one-step transition probability, $p_S(i,j)$, is the probability of transitioning from one state (i) to another (j) in a single step, that can be estimated by*

$$p_S(i,j) = \frac{N(i,j)}{\sum_{v \in S_S} N(i,v)}. \tag{1}$$

$N(i,j)$ *is the number of transitions from state i to j. All transition probabilities can be listed in a matrix as $P_S = \{p_S(i,j), i,j \in S_S\}$.*

**Definition 4 ($T_S$).** *The transition time from state i to j is a random variable with probability density $t_S(d;i,j)$ with d as the sojourn time of a specific transition between i and j. $T_S = \{t_S(d;i,j), i,j \in S_S\}$.*

*3.3. Markov Decision Process (MDP)*

The MDP models sequential decisions in a stochastic environment that can be represented by the five-tuple $\langle S_D, A_D, P_D, R_D, \Pi_D \rangle$ referring to the state space, action space, transition probability matrix, reward matrix and action selection policy.

**Definition 5 ($S_D$).** *The state space is a set of states that is used to describe a system/problem.*

**Definition 6 ($A_D$).** *The action space is a collection of actions that is able to control the system. Considering a queueing system, actions can be (1) 'open a counter', (2) 'maintain the current situation' and (3) 'close a counter', dependent on the current state, for example, the number of people in the queue.*

**Definition 7 ($P_D$).** *The transition probability from the current state i to the next state j under the impact of action a is denoted as $p_D(i,a,j)$. $P_D = \{p_D(i,a,j), i,j \in S_D, a \in A_D\}$.*

$$p_D(i,a,j) = \frac{N(i,a,j)}{\sum_{v \in S_D} N(i,a,v)}. \tag{2}$$

$N(i, a, j)$ *is the number of transitions from state i to j under the influence of action a.*

**Definition 8 ($R_D$).** *The immediate reward (positive or negative) $r_D(i, a)$ is obtained when the Markov chain stays at state i and makes action a.* $R_D = \{r_D(i, a), i \in S_D, a \in A_D\}$.

**Definition 9 ($\pi_D$).** *A policy at state i, denoted as $\pi_D(i)$, is a mapping from $S_D$ to $A_D$. That is, selecting the action that has the highest long-term reward.*

*3.4. Gamma Mixture Model*

**Definition 10** (gamma distribution)**.** *The gamma distribution is a two-parameter (non-negative shape z and non-negative rate $\lambda$) family of continuous probability distributions, which is widely used for modelling non-negative distributions. The density function is*

$$g(x|z, \lambda) = \frac{\lambda^z}{\Gamma(z)} x^{z-1} e^{-\lambda x}, x \geq 0 \tag{3}$$

*where $\Gamma(z)$ is the gamma function that equals $(z-1)!$ for positive integer z.*

**Definition 11** (gamma mixture model, ga-mm)**.** *The ga-mm is a mixture of multiple gammas with mixing proportions that sum to one. It is extremely flexible to fit any shape of positive continuous distributions. The density of a ga-mm is*

$$f(x|\Theta) = \sum_{i=1}^{m} \alpha_i g(x|z_i, \lambda_i) \tag{4}$$

*where $\Theta = (\alpha_1, \cdots, \alpha_m, z_1, \cdots, z_m, \lambda_1, \cdots, \lambda_m)$. $\alpha_i, z_i$ and $\lambda_i, i = 1, \cdots, m$ are the mixing proportion, shape and rate parameter of the ith gamma component, respectively.*

In the proposed HMDP method, gamma mixture models are integrated into the SMP model to characterize customers' duration data. The SMP is then applied as the bottom layer of the hierarchical MM. The top layer of the hierarchical MM is a high-level representation of customer patterns, derived from K-means. The detection/adaptation of concept drifts in customer transitions and their duration is then handled by an MDP.

**4. Methodology**

In this section, we introduce the proposed approach, called the hierarchical Markov model-based Markov decision process (HMDP). Using the hospital billing data as an example (as demonstrated in Table 1 and Figure 1), an illustration of the main logic structure of the method is given in Figure 2.

Given the old and new data/event log, we first represent customer behaviour by using an SMP separately. Customer transition behaviour is characterized using discrete-time transition probabilities, and the duration between every activity pair is modelled using gamma mixture models (ga-mm). We thereafter calculate the mean of these gamma mixture models and create a duration matrix that has the same size as the transition matrix, with each element referring to the average customer time spent on a specific transition. The transition and duration matrices are the low-level representation of the data with continuous elements. Consequently, the representation has uncountable variations, causing challenges in learning the MDP Q-table [39]. We, therefore, extract a high-level customer representation/pattern using the K-means technique [46]. Specifically, we combine the transition and duration matrices as a row vector and use it as the input of K-means. The output clustering centre is its high-level representation that has a categorical value. The high-level pattern of the old and new data contains information about the business environment evolution, and we combine them as the state of the MDP to make actions, detect concept drift and update the SMP model. Details of the approach are given in the following sub-sections.

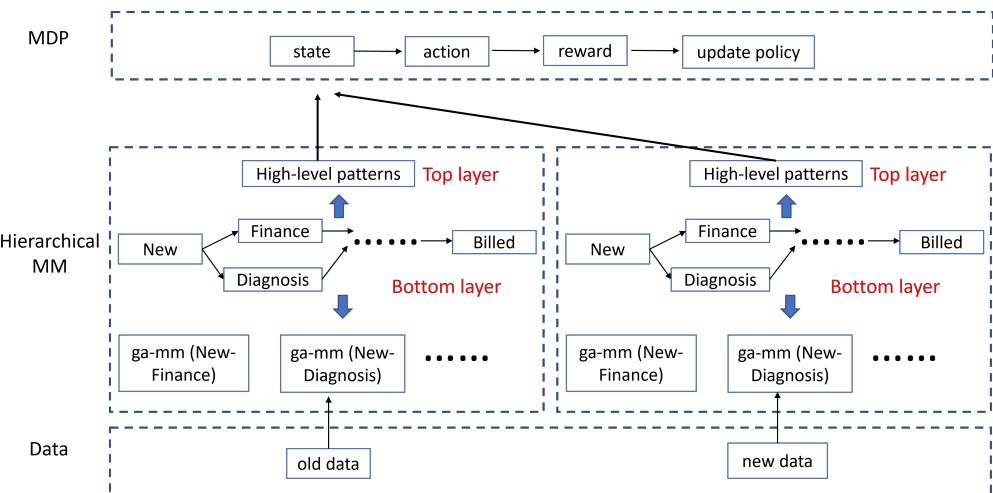

**Figure 2.** The flow chart of the HMDP approach.

### 4.1. Bottom Layer of the Hierarchical MM

The bottom layer of the hierarchical MM is an SMP with the duration estimated by ga-mm. As discussed in Section 3.2, $S_S$ is all the unique activities in a business process and $P_S$ covers the transition probabilities between each state transition. For example, in the hospital billing scenario, $S_S = \{New, FIN, ChangeDiagnosis, CodeOK, \cdots, Billed\}$. The time spent between each state transition from $i$ to $j$, $i, j \in S_S$, is fitted by a gamma mixture model with the number of mixing components optimized by the Bayesian information criterion (BIC) [47]. For example, we increase the number of mixing components from 1 to 10, and the ga-mm with the lowest BIC is selected as the best model. The Nelder–Mead algorithm [48] is used to compute the maximum likelihood estimator. The transition matrix ($P_S$) and gamma mixture models are the low-level representation of the customer behaviour behind the data. We then generate a matrix $G_S = g_S(i, j), i, j \in S_S$, consisting of the average duration for all state transitions, which is used for obtaining the top layer representation. $g_S(i, j)$ refers to the average transition time between state $i$ and $j$ and can be calculated using Equation (5).

$$g_S(i, j) = \sum_{v=1}^{m} \frac{\alpha_v \times z_v}{\lambda_v}.$$ (5)

$m$ is the number of mixing components. $\alpha_v$, $z_v$ and $\lambda_v$ are the mixing proportion, shape and rate parameter of the $v$th gamma component.

### 4.2. Top Layer of the Hierarchical MM

Using the bottom layer, i.e., the SMP, is promising for monitoring customer transition and duration behaviour, where a high-level representation is more efficient and effective for customer understanding and MDP learning. We first combine the transition ($P_S$) and duration ($G_S$) matrices as a row vector and use it as the input of the K-means model. The K-means output clustering centre is the high-level representation/pattern. Considering an event log originating from the hospital billing process that has eleven main activities. The size of the transition and duration matrix is $11 \times 11$, and therefore, the size of K-means input is $11 \times 11 \times 2 = 242$, and the output size is one. In other words, the high-level pattern is one-dimensional, and its value ranges from one to the number of predefined clusters. Furthermore, we can use two K-means models, respectively, for the transition and duration matrix, and in this case, the high-level pattern is two-dimensional.

We say it is a hierarchical MM because it has two layers—a bottom layer modelling low-level customer activities and duration and a top layer representing the high-level customer behaviour. The top layer, similar to hidden Markov models, includes a set of

hidden states dependent on the bottom later. Compared to hidden Markov models, which are commonly optimized using the maximum likelihood estimation, the top layer of our approach is obtained by K-means. In hidden Markov models, the hidden states are the bottom layer used to generate top layer customer behaviours, while in this paper, the hidden states are the top layer as a high-level representation of bottom layer customer behaviour.

*4.3. Model Management Using MDP*

We use tuple $\langle P_{SO}, G_{SO} \rangle$ and $\langle P_{SN}, G_{SN} \rangle$ to represent the transition and duration matrices of the old and new data, respectively. Furthermore, we use $H_O$ and $H_N$ as their high-level representation. We use tuple $\langle H_O, H_N \rangle$ as the MDP state that contains information about the environment evolution. Naturally, we say there is no concept drift if the old and new data have the same high-level representation, i.e., $H_O = H_N$. Otherwise, there is a high possibility for concept drift. During the MDP learning stage, depending on the current state, one of the actions is selected to update the current/old SMP model. The reward of the selected action is then calculated to update the decision-making policy. During the MDP testing stage, given the current state, i.e., the high-level environment representation derived from the old and new data, we select the action that achieves the highest long-term reward learned in the training process.

In the scenario that we use two K-means models, respectively, for the transition and duration matrix to have a two-dimensional high-level representation, actions can be made individually for transition probabilities and gamma mixture models, leading to nine actions in total. Consequently, the actions are 'keep_keep', 'keep_adapt', 'keep_retrain', 'adapt_keep', ⋯ and 'retrain_retrain'. For example, action 'keep_retrain' means that we maintain the current transition probabilities but retrain duration data into gamma mixture models. On the other hand, if the high-level representation is obtained using one K-means model, we have three actions: keep, adapt or retrain both the transition probabilities and gamma mixture models. In other words, the three actions are 'keep_keep', 'adapt_adapt' and 'retrain_retrain'.

Regarding actions for transition probabilities, if action 'keep' is selected, we consider there is no drift in the process control-flow and therefore keep the current transition probability matrix ($P_{SO}$) unchanged, i.e., $P_{SO}^{k+1} = P_{SO}^k$. $k$ and $k + 1$ refer to the transition probabilities before and after updating. If action 'adapt' is selected, we consider there is a gradual drift, and therefore, we update the model using the observed new data. That is, $P_{SO}^{k+1} = \frac{P_{SO}^k + P_{SN}^k}{2}$. We consider there is a sudden drift if action 'retrain' is selected and the current transition matrix is replaced by the new data. That is $P_{SO}^{k+1} = P_{SN}^k$. We denote the current gamma mixture models and the gamma mixture models derived from the new data batch as $ga\text{-}mm_O$ and $ga\text{-}mm_N$, respectively. For duration actions, if action 'keep' is selected, we keep the current mixture model unchanged, i.e., $ga\text{-}mm_O^{k+1} = ga\text{-}mm_O^k$. As a result, the low-level duration matrix is the same, i.e., $G_{SO}^{k+1} = G_{SO}^k$. If action 'adapt' is selected, we merge the old and new data to refit gamma mixture models. Otherwise, if action 'retrain' is selected, we refit the gamma mixture models using the new observation data.

The immediate reward of selecting action 'a' at state 's' ($\langle H_O, H_N \rangle$) is calculated as Equation (6). It has three components: (1) the accuracy of updated transition probabilities, (2) the goodness of fit of updated gamma mixture models and (3) a penalty for using a specific action. For the first component, we use the multinomial test [49] to measure the difference in each state transition between the updated transition probabilities and transition probabilities from the new data batch. In other words, the multinomial test is applied between each row of the updated transition matrix and the new data transition matrix. Consequently, there is a collection of independent multinomial test statistics and then the Fisher's method—a technique for detecting the difference in combining probabilities [50] is used with its *p*-value as the final performance measure. Therefore, Fisher's method measures transition probability changes in the entire process environment. It is straightforward that the better the selected action is, the better transition representation

we will have for the new data batch leading to a higher *p*-value. The second component evaluates the goodness of fit of the updated gamma mixture models for the new duration data. Every gamma mixture model is compared using the Kolmogorov–Smirnov (KS) test, and the Fisher's test is used for the multiple test problem. For the third component, we give no penalty to action 'keep', a mild penalty to action 'adapt' and a severe penalty for action 'retrain'. The reason for introducing the penalty mechanism is that we prefer to keep the current SMP model unchanged to save computational resources [16]. That is, action 'keep' will be in preference if 'adapt' and 'retrain' are not significantly better. In other words, without the penalty mechanism, the MDP agent will retrain the transition matrix and gamma mixture models at every decision point to receive the highest reward, which is not as we expected.

$$
\begin{aligned}
r_D(s,a) = \ & Fisher(Mult(\boldsymbol{P}_{SO}^{k+1}, \boldsymbol{P}_{SN}^{k})) \\
& + Fisher(KS(\boldsymbol{ga\text{-}mm}_O^{k+1}, \boldsymbol{ga\text{-}mm}_N^{k})) \\
& - penalty(a).
\end{aligned}
\tag{6}
$$

According to the Bellman optimality equation [39], the value of a state-action pair in MDP, denoted as $Q(s,a)$, can be written in a recursive fashion, i.e.,

$$
Q^{k+1}(s,a) = (1-\beta)Q^k(s,a) + \beta \sum_{j \in \boldsymbol{S_D}} p_D(s,a,j)[r_D(s,a) + \max_{b \in \boldsymbol{A_D}} Q^k(j,b)].
\tag{7}
$$

That is, we consider the *Q*-value optimization from two perspectives: a probability of $\beta$ for obtaining the future maximum reward and a probability of $1-\beta$ for preserving the current *Q*-value. In other words, $\beta$ is a trade-off between the exploration of a high reward and the preservation of historical knowledge. $p_D(s,a,j)$ is the transition probability that at state 's' via action 'a', the next state will shift to state *j*. It can be calculated as discussed in Definition 3. The policy, therefore, can be updated by,

$$
\pi_D(s)^{k+1} = \arg \max_{a \in \boldsymbol{A_D}} Q(s,a)^{k+1}.
\tag{8}
$$

Algorithm 1 details the learning/decision-making process of an MDP framework. We start to initialize the value and the policy for each state-action pair (lines 1–6). For example, we can give zero to all $Q(s,a)$ and a uniform distribution for $\pi(i)$. Following that, we generate the SMP model based on the old data (line 7) to obtain the transition matrix ($\boldsymbol{P}_{SO}^0$), gamma mixture models ($\boldsymbol{ga\text{-}mm}_O^0$) and the duration matrix ($\boldsymbol{G}_{SO}^0$). The high-level representation is then obtained using K-means (line 8). Define an empty array $SAJ$ to store the latest state, action, and next state pair for updating the transition probability in the MDP (line 9). While we observe a new data batch, whether it is in an online or offline format (line 10), the first step is to calculate its transition and duration matrix and gamma mixture models (line 11). The following step is to obtain the high-level representation $H_N$ (line 12). The high-level representation from the old and new data are then combined as the state in the MDP (line 13) and stored in $SAJ$ (line 14). Based on the current state 's', we select the best action 'a' that can reach the highest *Q*-value (line 18) and also store it in $SAJ$ (line 19). Notice that array $SAJ$ has two elements ('s' and 'a') after the first round iteration, and it will have three elements in the next iteration when we store the next state (line 15). Therefore, we can update the transition probability (line 16) and delete the first state-action pair (line 17). In this way, we can continuously update the transition matrix. According to the selected action, we can update the process presentation, i.e., the SMP model (line 20), calculate the immediate reward (line 21) and update the Q-factor and the policy (lines 22–23). Finally, we replace the previous SMP representation (*k*th iteration) using the newly updated representation ($k+1$th iteration) and wait for the next new data batch (lines 24–25).

---

**Algorithm 1** MDP policy iteration

---

**Input:** $S_D$: the state space,
$\quad\quad$ $A_D$: the action space,
$\quad\quad$ $P_D = \{p_D(i, a, j)\}$: the transition probability matrix,
$\quad\quad$ $\beta$: the learning rate.
**Output:** $\pi$: the policy.
1: **for** $i \in S_D$:
2: $\quad$ **for** $a \in A_D$:
3: $\quad\quad$ Initialize $Q^0(i, a) \in \mathbb{R}$
4: $\quad\quad$ Initialize $\pi^0(i) \in A_D$
5: $\quad$ **end for**
6: **end for**
7: Obtain $P_{SO}^0$, $ga\text{-}mm_O^0$, $G_{SO}^0$ from the old data.
8: Pass $P_{SO}^0$ and $G_{SO}^0$ into K-means to obtain $H_O^0$.
9: Let $SAJ = [\,]$, $k = 0$.
10: **while** A new data batch:
11: $\quad$ Obtain $P_{SN}^k$, $ga\text{-}mm_N^k$, $G_{SN}^k$.
12: $\quad$ Using K-means to obtain $H_N^k$.
13: $\quad$ Obtain MDP state $s = \langle H_O^k, H_N^k \rangle$.
14: $\quad$ Put $s$ into $SAJ$ for updating $P_D$.
15: $\quad$ **if** $SAJ$ has three elements:
16: $\quad\quad$ Update $P_D$ using Equation (2).
17: $\quad\quad$ Remove the first two elements.
18: $\quad$ Obtain action $a = \arg\max_{a \in A_D} Q(s, a)$.
19: $\quad$ Put $a$ into $SAJ$.
20: $\quad$ Update SMP, obtain $P_{SO}^{k+1}$, $ga\text{-}mm_O^{k+1}$, $G_{SO}^{k+1}$.
21: $\quad$ Calculate reward $r_D(s, a)$ using Equation (6).
22: $\quad$ Update the Q-factor using Equation (7).
23: $\quad$ Update $\pi_D(s)$ using Equation (8).
24: $\quad$ Update $H_O^{k+1}$ using $P_{SO}^{k+1}$ and $G_{SO}^{k+1}$.
25: $\quad$ $k = k + 1$.

---

## 5. Evaluation on Artificial Data

This section applies the HMDP approach to 68 artificial datasets referring to loan applications based on Maaradji et al.'s work [26]. These datasets have various characteristics, follow different process structures and contain sudden and gradual concept drift in customer transitions and their duration.

### 5.1. Datasets

#### 5.1.1. Sudden and Gradual Control-Flow Drift Data

By modifying a textbook example of a loan application process [51], demonstrated as the 'base' model in Figure 3a, Maaradji et al. [26] created 18 alternatives (e.g., 'cf' in Figure 3b) indicating control-flow changes in loops, parallel and alternative branches. Loan applications start from 'Check form' and end in one of the green activities. Process activities in blue boxes represent a loop structure. Thereafter, Maaradji et al. generated a set of event logs for each of the loan application models and then combined the 'base' model with each of the 18 'altered' models separately for simulating control-flow sudden drift, as demonstrated in Figure 4. By control-flow sudden drift, we mean abrupt variations in customer behaviour through the process due to changes from the process management or the user side.

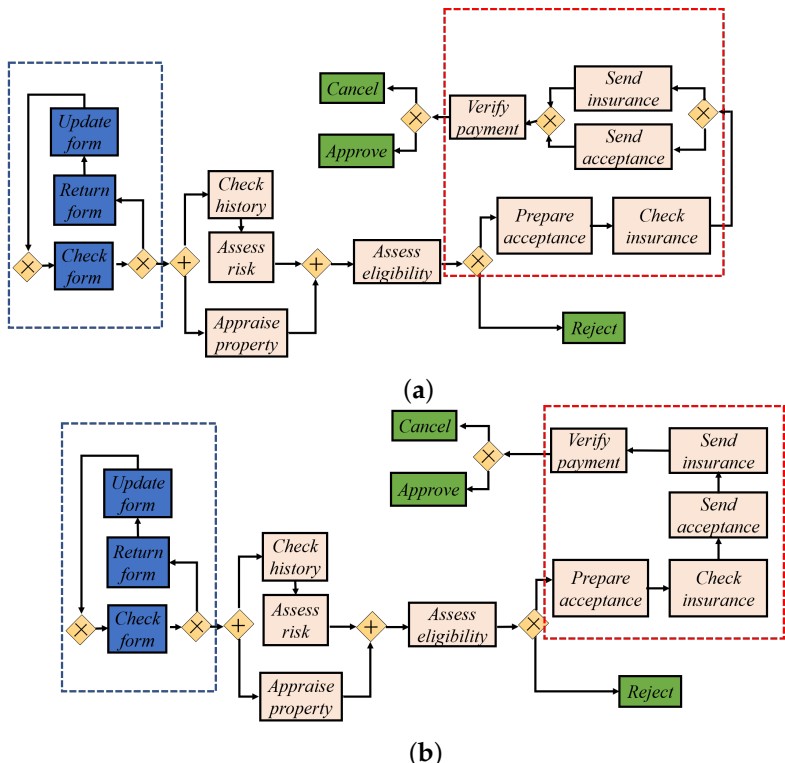

**(a)**

**(b)**

**Figure 3.** The base model and one of its alternative versions. (**a**) base model; (**b**) altered model ('cf').

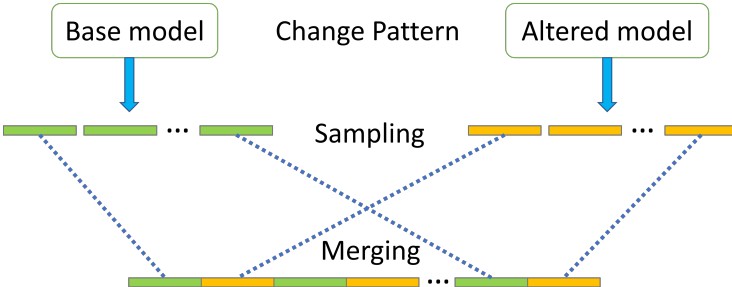

**Figure 4.** Log generation for sudden drifts.

In this paper, we generated another 18 gradual control-flow changes, as illustrated in Figure 5. There exist two types of special areas (the red and blue bars) in which process instances are generated by both the 'base' and the 'altered' model. That is, instances in the green bars were completely created by the 'base' model, and then during the red bars, instances were generated based on the probabilities shown in Figure 6a. Generally speaking, the fading of the old process model and the taking over of the new model happen linearly. Regarding red bars, the 'base' model is gradually replaced by the 'altered' model, and before the crossover point (in Figure 6a), samples are more likely to be sampled from the 'base' model, and vice versa for instances after the crossover point. Similarly, instances in orange bars are totally from the 'altered' model, and during the blue bars, the 'altered' model is gradually taken over by the 'base' model.

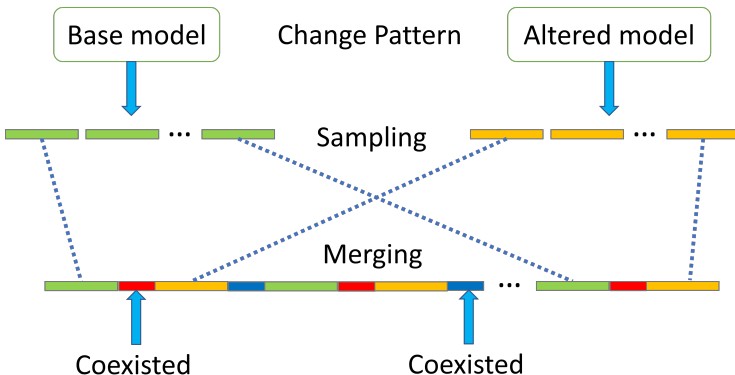

**Figure 5.** Log generation for gradual drifts.

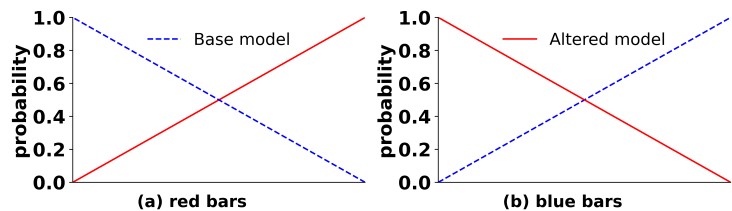

**Figure 6.** Probability of sampling during the periods of gradual drift.

### 5.1.2. Sudden and Gradual Duration Drift Data

To generate datasets containing duration drift, we first reorganized the 'base' model by maintaining its control-flow structure but simulating all state transitions to follow the same distribution: a mixture of an exponential distribution (with rate and mean equal one) and a gamma distribution (with shape and mean equal five), weighted by a 50% probability. Thereafter, we generated 16 'altered' models by altering the duration distribution of the three transitions in the blue box in Figure 3. Specifically, eight of them were generated by increasing the gamma shape parameter from 5 to 21 with an increment of two, maintaining the exponential distribution and the mixing proportion unchanged. The other eight alternative models were created by tuning the mixing proportion from 10% to 90% with an increment of 10%, instead of the original 50% setup. Following that, we generated event logs for all the 'base' and 'altered' models and created datasets involving sudden and gradual drifts using the same strategy (as shown in Figures 4 and 5). Finally, we created 16 sudden duration drift logs and 16 gradual duration drift logs.

### 5.2. Experimental Setup

In a sudden drift context, we sampled 2000 process instances for each of the 'base' (green) and 'altered' (orange) bars. We generated 20 (10 green and 10 orange) bars for each of the 34 (18 transition drift and 16 duration drift) sudden datasets, leading to 40,000 process instances in total in each dataset. For gradual drift, the green and orange bars still have 2000 process instances, but during the gradual drift period, i.e., the red and blue bars, 1000 instances were generated. Consequently, each gradual drift dataset has 120,000 process instances.

This section validates the proposed HMDP approach in an offline manner by splitting datasets into equal-sized populations (800 process instances each). The first half of population is used to optimize the MDP, i.e., to optimize the policy $\pi$. The remaining populations are used to evaluate the performance. To start with, the first and second populations are used as the old and new data, respectively, to obtain their low- and high-level representation. The high-level representations are then combined as the state for action selection, as well as the update of transition probabilities and gamma mixture models. In the next iteration, the third population is considered as the new data and the learning process continues until the last population. The number of clusters in K-means is predefined as three to represent three possible scenarios: (1) all process instances are from the 'base' model,

(2) all process instances are from the 'altered' model, (3) process instances are generated by both the models. We set the action penalty as zero for action 'keep' and tested the performance under different penalties for action 'adapt' and 'retrain'. Furthermore, we compared the HMDP action selection policy with four other policies: 'Always keep' (i.e., action 'keep_keep'), 'Always adapt', 'Always retrain' and 'Random'. By using action 'Random', one of the three actions is selected under an equal probability for both the transition probabilities and gamma mixture models.

*5.3. Experimental Results*

The HMDP approach was validated under different penalties. That is, the penalty for action 'retrain' is increased from zero to one with an increment of 0.1, associated with a 50% discount for 'adapt'. For example, under a penalty of 0.6 for 'retrain', the penalty is 0.3 for 'adapt'. Tables 2 and 3 demonstrate the average reward of HMDP and other action selection policies under all the above-mentioned datasets. Column 'penalty' represents the penalty for action 'retrain', and the best policy is highlighted in bold type separately for sudden and gradual drift. Without action penalty, the average reward ranges from zero to two because the *p*-value of Fisher's method for both transitions and duration ranges from 0 to 1. Under action penalty one, there can be a negative penalty ranging from $-2$ ('retrain_retrain') to 0 ('keep_keep') and, therefore, the average reward can be negative but within the period between $-1$ (action 'adapt_adapt' with both Fisher's *p*-value equal to 0) and 2 (action 'keep_keep' with Fisher's *p*-value as 1).

**Table 2.** The average reward of the 18 sudden and gradual transition drifts.

| Penalty | Sudden Drifts | | | | | Gradual Drifts | | | | |
|---|---|---|---|---|---|---|---|---|---|---|
| | Keep | Adapt | Retrain | Random | HMDP | Keep | Adapt | Retrain | Random | HMDP |
| 0 | 1.160 | 0.884 | **2.000** | 1.377 | 1.967 | 0.726 | 0.954 | **2.000** | 1.321 | 1.495 |
| 0.1 | 0.783 | 0.692 | **1.800** | 1.155 | 1.697 | 0.714 | 0.847 | **1.800** | 1.200 | 1.787 |
| 0.2 | 0.770 | 0.619 | **1.600** | 1.044 | 1.529 | 0.739 | 0.747 | **1.600** | 1.117 | 1.620 |
| 0.3 | 0.793 | 0.507 | **1.400** | 0.916 | 1.373 | 0.731 | 0.664 | 1.400 | 1.004 | **1.439** |
| 0.4 | 0.780 | 0.391 | 1.200 | 0.828 | **1.221** | 0.712 | 0.544 | 1.200 | 0.881 | **1.262** |
| 0.5 | 0.773 | 0.299 | 1.000 | 0.753 | **1.064** | 0.721 | 0.444 | 1.000 | 0.812 | **1.096** |
| 0.6 | 0.786 | 0.205 | 0.800 | 0.632 | **0.888** | 0.727 | 0.347 | 0.800 | 0.728 | **0.928** |
| 0.7 | **0.786** | 0.095 | 0.600 | 0.529 | 0.740 | 0.734 | 0.261 | 0.600 | 0.619 | **0.750** |
| 0.8 | **0.788** | 0.005 | 0.400 | 0.476 | 0.595 | **0.740** | 0.156 | 0.400 | 0.497 | 0.573 |
| 0.9 | **0.796** | $-0.099$ | 0.200 | 0.347 | 0.431 | **0.724** | 0.038 | 0.200 | 0.397 | 0.405 |
| 1.0 | **0.778** | $-0.204$ | 0.000 | 0.260 | 0.276 | **0.739** | $-0.049$ | 0.000 | 0.313 | 0.229 |

**Table 3.** The average reward of the 16 sudden and gradual duration drifts.

| Penalty | Sudden Drifts | | | | | Gradual Drifts | | | | |
|---|---|---|---|---|---|---|---|---|---|---|
| | Keep | Adapt | Retrain | Random | HMDP | Keep | Adapt | Retrain | Random | HMDP |
| 0 | 1.042 | 1.196 | **2.000** | 1.339 | 1.741 | 1.051 | 1.244 | **2.000** | 1.384 | 1.873 |
| 0.1 | 1.041 | 1.097 | **1.800** | 1.279 | 1.650 | 1.056 | 1.114 | **1.800** | 1.246 | 1.725 |
| 0.2 | 1.036 | 0.988 | **1.600** | 1.158 | 1.538 | 1.049 | 0.993 | **1.600** | 1.172 | 1.564 |
| 0.3 | 1.058 | 0.886 | **1.400** | 1.052 | 1.398 | 1.043 | 0.901 | 1.400 | 1.062 | **1.402** |
| 0.4 | 1.051 | 0.795 | 1.200 | 0.961 | **1.269** | 1.061 | 0.792 | 1.200 | 0.972 | **1.285** |
| 0.5 | 1.045 | 0.691 | 1.000 | 0.872 | **1.165** | 1.051 | 0.700 | 1.000 | 0.870 | **1.135** |
| 0.6 | 1.046 | 0.610 | 0.800 | 0.745 | **1.058** | **1.040** | 0.625 | 0.800 | 0.751 | 0.965 |
| 0.7 | **1.039** | 0.494 | 0.600 | 0.666 | 0.934 | 1.057 | 0.499 | 0.600 | 0.655 | 0.767 |
| 0.8 | **1.045** | 0.400 | 0.400 | 0.604 | 0.748 | **1.052** | 0.414 | 0.400 | 0.558 | 0.659 |
| 0.9 | **1.043** | 0.286 | 0.200 | 0.469 | 0.599 | **1.047** | 0.310 | 0.200 | 0.480 | 0.497 |
| 1.0 | **1.047** | 0.216 | 0.000 | 0.349 | 0.306 | **1.058** | 0.199 | 0.000 | 0.375 | 0.327 |

As we expected, without a penalty, action 'retrain' can achieve 100% accuracy in modelling new data's transitions and durations because the bottom layer representation is updated using new data and then used to estimate the new data's goodness of fit. Consequently, the HMDP approach becomes invalid for detecting concept drift and manage model updating. On the contrary, using penalty one, Fisher's method can have 100%

accuracy with the *p*-value as one, but there is a negative penalty of −1, leading to zero reward at the end. There is no penalty for action 'keep', so it remains stable, with the reward fluctuating around one, indicating its ineffectiveness. Because all datasets solely contain one drift type, either in transitions or durations, the action 'keep' can achieve a *p*-value close to one for the unchanged part, i.e., it has a bottom line around one. In other words, the action 'keep' has an average reward of around one, indicating that it can not represent new data under concept drift scenarios. Action 'adapt' performs unsatisfactorily under all scenarios because it combines both the old and new data, and the mixed data can not precisely represent the environment. The HMDP approach can outperform the baseline policy 'random' and other policies under appropriate penalties, indicating its effectiveness. Basically, a penalty between 0.4 and 0.7 is suitable. In conclusion, the action 'retrain' concentrates on the goodness of fit of the process environment, requiring updating the bottom layer customer representation every time. Action 'keep' saves computational resources since it does nothing during the environment evolution and, consequently, is disabled to represent the environment. The HMDP approach benefits both the accuracy and efficiency in environment modelling by automatically retraining the model under concept drift and maintaining the model otherwise.

Using penalty 0.5 as an example, we illustrate the average reward for transition and duration drift in Figure 7. The first two subfigures refer to the average reward of the five action selection policies under sudden and gradual transition drift, respectively. Figure 7c,d relate to the average reward under duration and concept drift. The horizontal axis of Figure 7a,b is the 18 control-flow change datasets. In Figure 7c,d, the horizontal axis refers to the 16 duration drift patterns with the first eight elements as changes in mixing proportions and the latter eight elements as drift in the gamma shape parameter. The mean of the reward of each policy in Figure 7 is the value in Tables 2 and 3 with a penalty of 0.5.

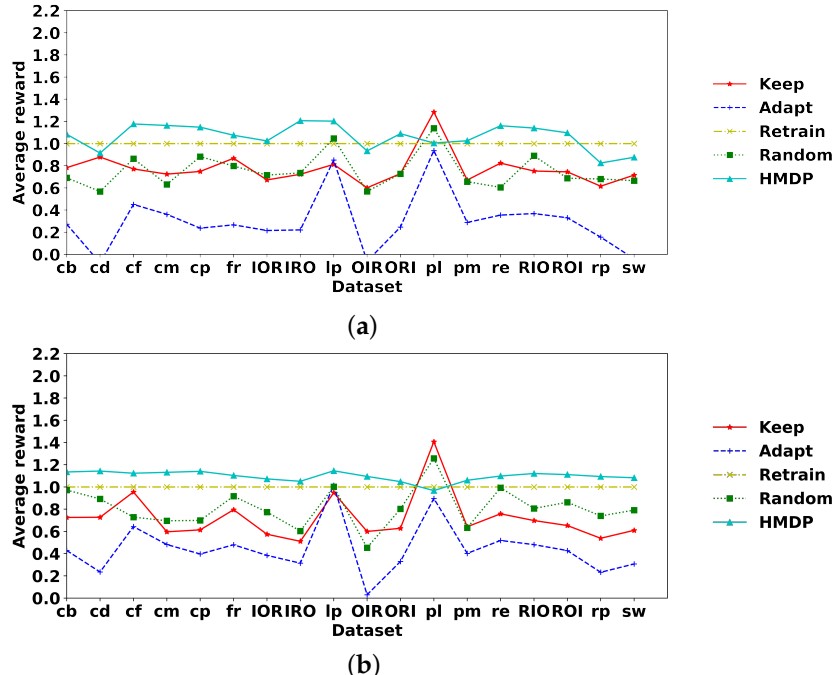

**Figure 7.** *Cont.*

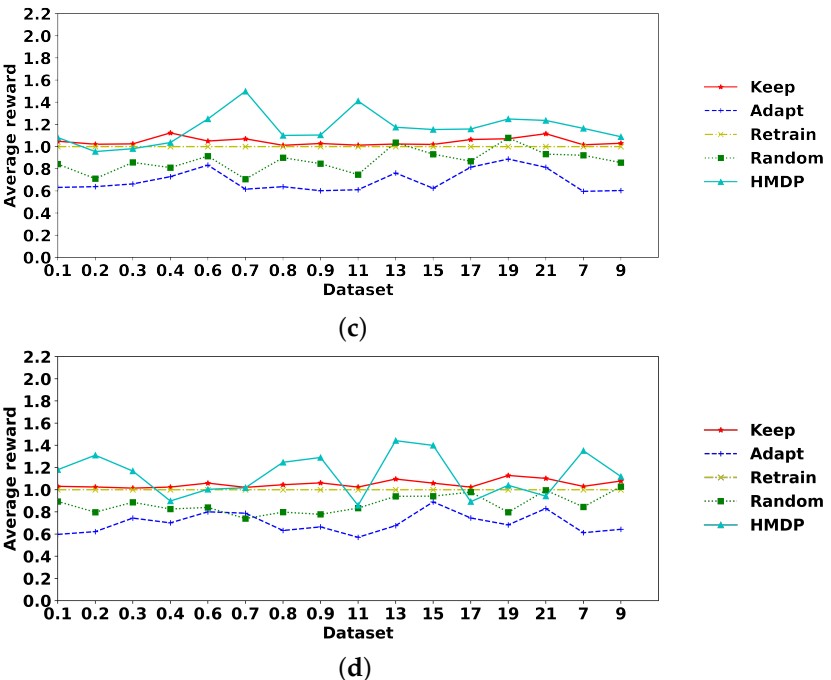

**Figure 7.** Performance measure under artificial datasets with a penalty of 0.5. (**a**) Sudden transition drift; (**b**) gradual transition drift; (**c**) sudden duration drift; (**d**) gradual duration drift.

## 6. Evaluation of the Real-World Hospital Billing Data

In this section, the HMDP approach is evaluated using a real-world hospital billing (HB) dataset [52]. The HB data were collected between December 2012 and January 2016, containing 18 process activities, 451,359 events and 100,000 cases [53]. There are eleven main operations, covering 99% process instances (98,515), with the discovered control-flow structure presented in Figure 1.

We validate the HMDP approach in an online learning scenario. That is, we consider a new data batch as arriving in time windows such as a month or a week. In other words, the customer behaviour and environment representation are updated on a daily or weekly basis. Consequently, the observed process instances can be incomplete, i.e., truncated or censored [54]; for example, a process instance started within the observation but has not finished at the end of the observation. In this case, we have no information for its following activities at the time point we make decisions for concept drift detection and model refinement. Furthermore, the number of process instances in a new data batch is changeable depending on the number of applications in that observation window. This is different to the experiments on the artificially created loan applications. In that offline learning scenario, we sample a fixed number of process instances (i.e., the number of process instances is the same in every data batch) that are complete with a starting and ending activity. We split the HB event log into two sets where the first one started from December 2012 to July 2014 as the training stage to optimize the MDP action selection policy and determine the penalty. During the testing stage, data between August 2014 and January 2016 are used for performance evaluation.

The performance of the five action selection policies is validated under different penalties on a weekly basis, as demonstrated in Figure 8. The HMDP approach performs the best under a penalty of 0.3 and 0.4, and therefore, we use 0.4 in the testing stage. The average reward of the five policies is demonstrated in Figure 9. The horizontal axis refers to the arrival of new data, and the vertical axis represents the reward for each new data. The average reward of each policy is illustrated in the legend area. In general, the HMDP approach can achieve the best performance with an average reward of 1.36, followed by the policy of 'random', 'retrain' 'keep' and 'adapt'. The HMDP is more stable than other methods with mild fluctuation.

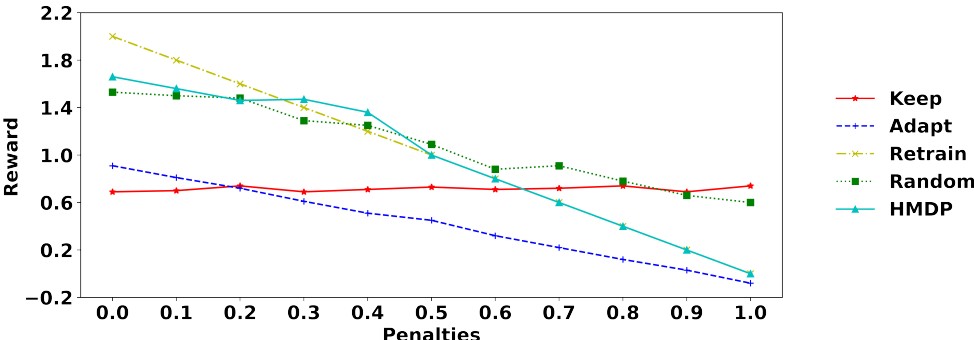

**Figure 8.** The average reward under different penalties.

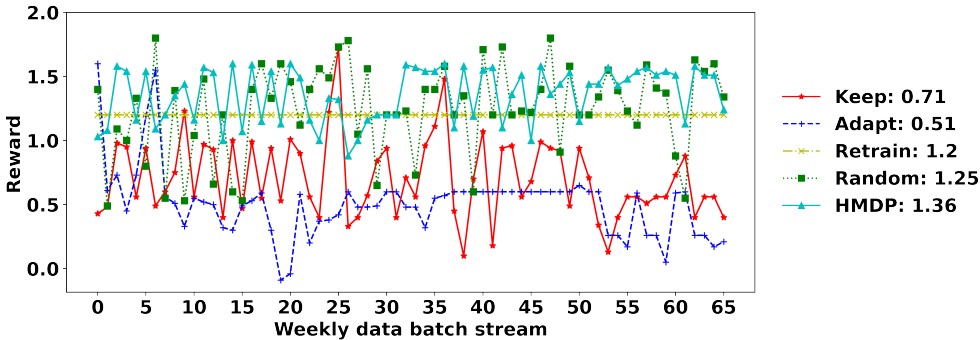

**Figure 9.** The average reward under different policies.

## 7. Conclusions

This paper proposes a technique, namely the hierarchical Markov model-based Markov decision process (HMDP), for business process concept drift detection and customer behaviour representation refinement. A hierarchical MM is generated for modelling customer transition and duration behaviour into a low- and high-level representation. The bottom layer is a semi-Markov process with customer discrete-time transition probabilities and gamma mixture model fitted duration. The top layer is a high-level representation of customer behaviour and the process environment, which was achieved using the K-means technique. The high-level representation of the old and new data are then combined as the MDP state for action selection and policy update. We validated the approach in an offline manner using 68 artificially created sudden and gradual drifts that occur in customer transitions and duration, as well as in an online setup on a real-world hospital billing log. Experiments demonstrate the effectiveness of the approach in drift detection and autonomous customer behaviour representation refinement.

In its current form, the top layer of the hierarchical MM is created using the K-means, while in the future, we are interested in learning the hidden state using maximum likelihood estimation to model the hierarchical Markov structure in a formal way. In the MDP, part of the HMDP approach, the long-term reward and action policy are updated using a Q-table strategy, which causes difficulties in applications under complex business process scenarios that have a large state and action space. To solve this problem, a possible direction for future work is to consider the use of deep neural networks for value estimation (for example, Deep Q-network) and action policy optimization. Furthermore, we are interested in applying the technique in process datasets with already known concept drift areas. The final direction is to apply the framework for modelling big data from British Telecom.

**Author Contributions:** Conceptualization, L.Y., S.M., M.D., K.B. and K.K.; methodology, L.Y., S.M., M.D. and K.B.; software, L.Y.; validation, L.Y.; formal analysis, L.Y.; investigation, L.Y.; resources, L.Y.; data curation, L.Y.; writing—original draft preparation, L.Y.; writing—review and editing, S.M., M.D. and K.B.; visualization, L.Y.; supervision, S.M., M.D., K.B. and K.K.; project administration, S.M.; funding acquisition, S.M. and K.K. All authors have read and agreed to the published version of the manuscript.

**Funding:** This research was funded by British Telecom and Invest Northern Ireland.

**Institutional Review Board Statement:** Not applicable.

**Data Availability Statement:** Publicly available datasets were analyzed in this study. This data can be found here: [https://data.4tu.nl/articles/dataset/Business_Process_Drift/12712436 (accessed on 14 March 2022)] and [https://data.4tu.nl/articles/dataset/Hospital_Billing_-_Event_Log/12705113 (accessed on 14 March 2022)].

**Acknowledgments:** This research is supported by BTIIC (the British Telecom Ireland Innovation Centre), funded by British Telecom and Invest Northern Ireland.

**Conflicts of Interest:** The authors declare no conflict of interest. The funders had no role in the design of the study; in the collection, analyses, or interpretation of data; in the writing of the manuscript, or in the decision to publish the results.

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
