# Peer review of "Detecting and Responding to Concept Drift in Business Processes"

_algorithms, doi:10.3390/a15050174_

Round 1

Reviewer 1 Report

This article “Detecting and responding to concept drift in business processes”, the authors model response to concept drift as a sequential decision making problem by combing a hierarchical Markov model and a Markov decision process (MDP). The approach can detect concept drift, retrain the model and update customer profiles automatically.

This is a good application model in practice and provides algorithms, which is helpful for practical application, some suggestions are as follows:

  1. It is recommended to draw a big picture to illustrate the key points of this article.
  2. The authors validated the proposed approach on 68 artificial datasets…, it is recommended that the authors provided more case studies, rather than just verifying through 68 artificial datasets.
  3. The authors should explain from the research findings why “This paper models customer transitions and their time spent behaviour as a two-layer hierarchical Markov model.”?
  4. Please explain how the definition of Section 3 is applied in Section 4 – Methodology to introduce the proposed approach, called the hierarchical Markov model-based Markov decision process (HMDP).

Reviewer 2 Report

The article presents a proposal for a novel technique for business process concept drift detection based on the use of hierarchical Markov models.

The proposal is interesting and no inconsistencies have been detected in its description and development. The model has been validated using both synthetic data and data obtained from a real (albeit relatively simple) business process.

The proposal is meritorious and the manuscript is well structured. However, it is recommended to improve the description of business process concept drift detection models that can be found in the scientific literature (section 2), as well as, mainly, to include a reasoned discussion of the improvements that the proposed technique entails compared to the existing proposals in the literature.
